# Diversity of Freshwater Calanoid Copepods (Crustacea: Copepoda: Calanoida) in North-Eastern China

**Ruirui Ding** [1,2], **Le Liu** [1,2], **Shusen Shu** [3,4], **Yun Li** [1] **and Feizhou Chen** [1,2,*]

1. State Key Laboratory of Lake Science and Environment, Nanjing Institute of Geography and Limnology, Chinese Academy of Sciences, Nanjing 210008, China; realdrr@163.com (R.D.); liule181@mails.ucas.ac.cn (L.L.); liyun@niglas.ac.cn (Y.L.)
2. University of Chinese Academy of Sciences, Beijing 100049, China
3. State Key Laboratory of Genetic Resources and Evolution, Kunming Institute of Zoology, Chinese Academy of Sciences, Kunming 650223, China; shuss@mail.kiz.ac.cn
4. Southeast Asia Biodiversity Research Institute, Chinese Academy of Sciences, Nay Pyi Taw 05282, Myanmar
* Correspondence: feizhch@niglas.ac.cn

**Abstract:** The distribution and diversity of calanoid copepods were investigated using samples collected from 37 lakes in North-eastern (NE) China in 2019. A total of 10 calanoid copepods belonging to eight genera and three families were identified. Among them, *Heterocope soldatovi* Rylov, 1922 was recorded for the first time in China. Species from the family Centropagidae were more widespread than those from the families Temoridae and Diaptomidae. *Sinocalanus doerrii* (Brehm, 1909), previously not recorded in NE China, is now widespread alongside *Boeckella triarticulata* (Thomson G.M., 1883), and the latter species is also prevalent in that region. Canonical correspondence analysis (CCA) and redundancy analysis (RDA) revealed that calanoid copepods were significantly correlated with total phosphorus (TP), total nitrogen, conductivity, nitrate nitrogen, altitude, and dissolved organic carbon. TP was the most important environmental variable that impacted the distribution of calanoid copepods, including both fresh and saline–alkaline lakes. Integrating historical records, a total of 21 calanoid copepods were distributed in NE China, and we also gave comments on the ecology and distribution of these species.

**Keywords:** calanoid; distribution; diversity; environmental variables; North-eastern China

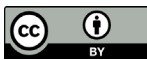

## 1. Introduction

Copepods are tiny aquatic planktonic organisms that are widely distributed in various inland water bodies, such as lakes, ponds, rivers, and groundwater [1–3]. Freshwater copepods belong to three main orders: Calanoida, Cyclopoida, and Harpacticoida. Among these, calanoids are the dominant group in freshwater ecology and fisheries in the pelagic zones of lakes, estuaries, and ponds. To date, the family Diaptomidae is the most species-rich of the inland water calanoid copepods [4], with more than 440 species in 62 genera [5]. The freshwater Diaptomidae consists of about 56 species of 19 genera in China [6] and tends to have a restricted spatial distribution. Nowadays, 18 calanoid species have been reported from North-eastern (NE) China [7–10].

The distribution and diversity of calanoid copepods can be greatly affected by the local and regional environment. Temperature is a key abiotic factor affecting the distribution of calanoids [11–14]. It also affects the body size of calanoids, which tend to become smaller with increasing temperature [15,16]. As salinity increased, most calanoid species in the water decreased, leaving mostly brackish species [17,18]. Calanoids had a wide range of pH adaptations and were found in both acidic and alkaline waters, but cyclopoids were absent or had low abundance in acidic waters [19–21]. Lake productivity (trophic state) was a major determinant; the trophic state affected the size and quality of

food for calanoids to some extent, which in turn affected their growth and abundance [20,22,23]. Interactions between aquatic organisms could also have an impact on calanoids, such as fish predation, coexistence, etc. Due to their body size and swimming style, calanoids were more likely to be preyed upon by planktivorous fish. Fish predation could alter the life strategy and body size of calanoids, thereby altering their population density and community structure [24–29]. Thus, the biomass ratio of calanoids to cyclopoids could reflect the pressure of fish predation, with the ratio decreasing as predation increased [25,30].

The 37 lakes in this study, located in the north-east of China, were predominantly eutrophic [31]. Previous studies mostly focused on the zooplankton diversity of a particular lake, and the overall distribution and diversity of calanoid copepods in the north-east of China were not clear. In addition, although the taxonomic studies of calanoid copepods in this area have been continuously updated [32–34], the status of calanoid species diversity was not clear, and the species list of the region has never been systematically updated since [7]. In this paper, based on the literature data and our survey results, the new calanoid species list in NE China was given, and the taxonomy and distribution status of each species were evaluated. We also revealed the relationship between calanoid copepods and environmental variables.

## 2. Materials and Methods

### 2.1. Study Area

This study was conducted in NE China. We selected 37 lakes, which were located at latitudes from 41.75° N to 49.87° N and longitudes from 117.59° E to 134.28° E (Figure 1). The Siberian continental air mass is controlled in winter by a long glacial period, and the subtropical oceanic air mass is controlled in summer, which is warm and rainy. The region is a vast expanse of plains surrounded by mountains and crisscrossed by rivers and lakes. This area is one of the most dense lake regions in China [35]. The central plain area has a large area of the underground impermeable layer due to the low-lying terrain, and most of the surface water accumulates in lakes. Most of the lakes in this region are characterised by small areas and shallow, saline, alkaline, and temporary water. In contrast, the formation of lakes in the north-western and south-eastern mountains is generally associated with volcanic activity and tectonic movements, such as the mountain barrier lake (Lake Jingpo) and the tectonic trap lake (Lake Xingkai).

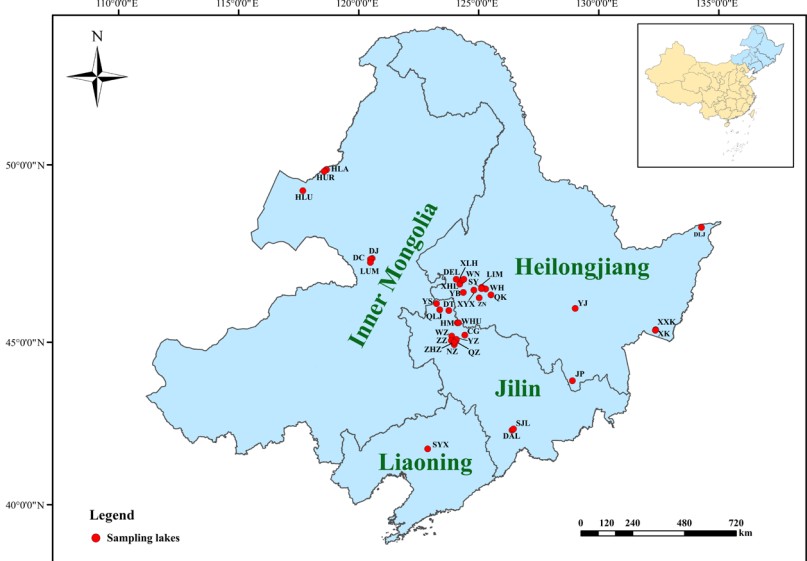

**Figure 1.** Geographical location of sampling lakes. Lakes shown are Wanghua (WH), Datun (DT), Liming (LM), Xihulu (XHL), Sayong (SY), Hulun (HLU), Chagan (CG), Yuebing (YB), Delong (DEL),

Dichi (DC), Dalong (DAL), Xiaoxingkai (XXK), Xingkai (XK), Dalijia (DLJ), Hala (HLA), Xiaolonghu (XLH), Yuejin (YJ), Qingken (QK), Sanjiaolong (SJL), Yangsha (YS), Qianliujia (QLJ), Dujuan (DJ), Luming (LUM), Nvzi (NZ), Zhenzi (ZZ), Hama (HM), Wangzi (WZ), Qianzi (QZ), Jingpo (JP), Xiuyixi (XYX), Woniu (WN), Shenyangxi (SYX), Zhongnei (ZN), Yezi (YZ), Zhouzi (ZHZ), Wanghuan (WHU), and Hure (HUR).

### 2.2. Sampling, Preservation, and Identification

Water temperature (WT), pH, salinity, and dissolved oxygen concentration (DO) were measured using a YSI 6600 (Yellow Springs, OH, USA). Secchi depth was determined using a Secchi disk. Mixed layers of water were collected by a 5 L Schindler sampler for chemical analyses. Total nitrogen (TN), ammonium nitrogen ($NH_4$-N), nitrate nitrogen ($NO_3$-N), total phosphate (TP), and dissolved organic carbon (DOC) were determined according to standard methods [36]. Chlorophyll *a* (Chl *a*) concentration was extracted with ethanol and measured using a UV-Vis spectrophotometer [37].

Due to the long frozen period in winter, copepod samples were sampled in July 2019, when the individual matured. A total of 111 samples were collected in 37 lakes, the 15 L depth mixture water was filtered through a 64 μm mesh net to collect calanoid copepods and concentrated to 50 mL, and samples were preserved in 4% formaldehyde solution for further analyses. In the laboratory, calanoid copepods were identified using the identification keys of Shen and Song [7] and Sheveleva et al., 2021 [34]. The animals were counted and measured under a Carl Zeiss optical microscope. The body lengths of calanoid copepods were measured for 20 individuals (if not enough, measure all of them) to estimate the dry biomass following the length–weight relationship [38,39].

The abbreviations used are as follows: A1: antennule; P1–P5: the first–fifth pair of legs; P5bsp1: the first segment of the basipodite of the fifth pair of legs; P5bsp2: the second segment of the basipodite of the fifth pair of legs; P5exp1: the first segment of the exopodite of the fifth pair of legs; and P5exp2: the second segment of the exopodite of the fifth pair of legs.

### 2.3. Data Analysis

The ten physicochemical indicators of 31 lakes with calanoid copepods were clustered in a Q-type system, standardised by Z scores, using the Euclidean distance and Group Average Linkage method [40,41]. We performed a standardised data exploration to detect outliers, then $log_{10}$ [x + 1] transformed before performing statistical analyses to reduce the variance in the physical–chemical data and to avoid violating normality assumptions. To identify differences in the environmental variables of each type of lake, a one-way ANOVA with a post hoc Tukey's honestly significant difference test was applied. Data were analyzed using the Canoco 5.0 program to explore correlations between calanoid species, habitats, and environmental variables and between species and environmental variables. Correlations between calanoids biomass and environmental variables were first examined in each habitat using detrended correspondence analysis (DCA), and all gradient lengths were less than 3. Redundancy analysis (RDA) was then chosen; otherwise, canonical correlation analysis (CCA) was chosen. The RDA and CCA procedures produce an ordination diagram in which habitats are represented by points and species and environmental variables are represented by vectors. Monte Carlo permutation tests were performed to test the statistical strengths of the eigenvalues of the ordination axes and species–environmental correlations. The minimum number of measured environmental variables that could account for the majority of the variance in the species data was determined by forward selection with permutation tests at $p < 0.05$. The frequency of species occurrence was calculated based on the cumulative number of all collected samples with calanoid species. All the tests and statistical analyses were performed using Excel, SPSS 23.0, Prism 9, and Canoco for Windows 5.0.

## 3. Results

### 3.1. Lake Chemistry and Cluster Analysis

No calanoids were found in six lakes in our survey, namely Lake Shenyangxi, Lake Zhongnei, Lake Yezi, Lake Zhouzi, Lake Wanghua, and Lake Hure. Based on the eleven physicochemical factors of the lakes (Table S1), the cluster analysis could classify the 31 lakes in which calanoids occurred into five types when the distance coefficient was 10 (Figure 2): Type 1: these lakes were mostly freshwater lakes, eutrophic, and salinity was 0–1.52 g/L; most of the lakes were located in the plain, and the size varied; Lake Dichi, Lake Dujuan, and Lake Luming were volcanic lakes and mountain barrier lakes (altitude above 800 a.s.l.), and salinity was less than 0.5 g/L, covered by aquatic plants. Type 2: these lakes were mostly hyposaline lakes with high organic matter (except Lake Qianzi), salinity was 0.5-4.0 ppt, and shallow lakes and typical alkaline lakes were present in the region. Type 3: this type consisted only of Lake Jingpo, which is the largest mountain barrier lake in China, a river-type lake, obviously disturbed by anthropogenic activities, with low mineralisation. Type 4: only one lake, Lake Xiuyixi, was included in this type; it is a hyposaline lake with high pH and organic matter and high concentrations of nitrogen and phosphorus nutrients. Type 5: only Lake Woniu was included; it is shallow with high mineralisation, salinity was 5.65 g/L, and the nutrient concentration in this lake was significantly higher than in the other four lake types ($p < 0.0001$).

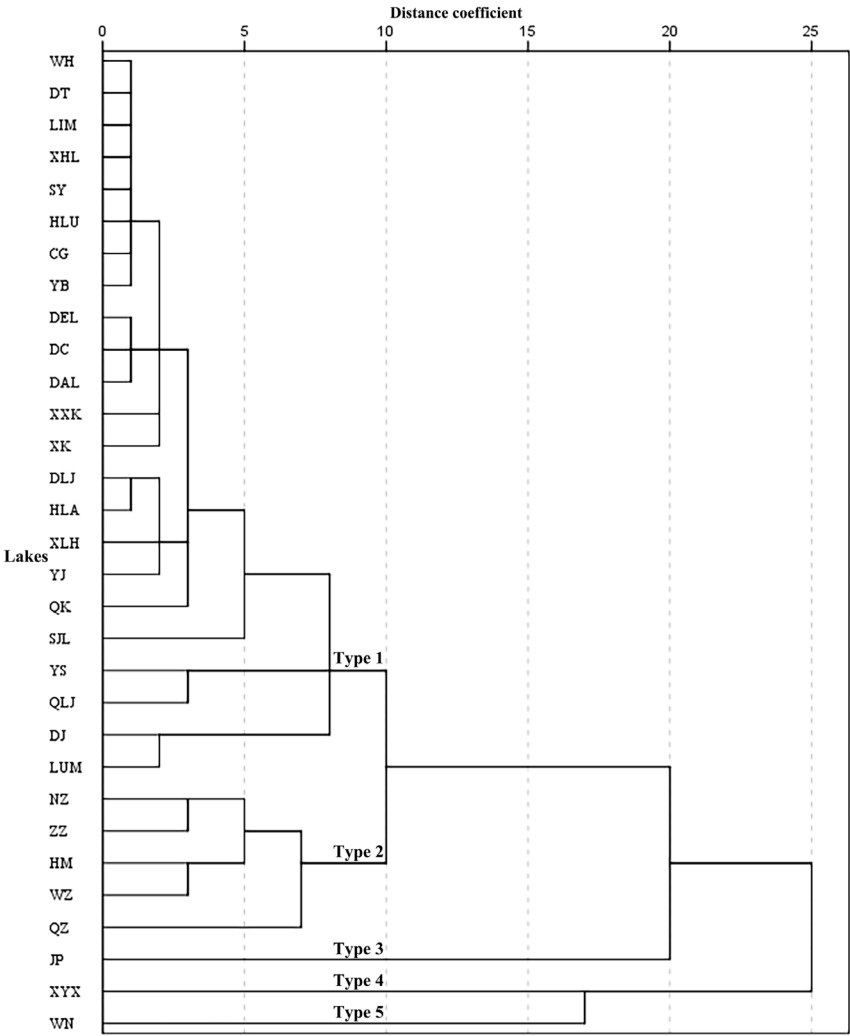

**Figure 2.** Cluster analysis based on Euclidean distance of physical–chemical factors in 31 lakes with calanoid occurrence.

The five types of lakes differ significantly in salinity, nitrogen, phosphorus nutrient content, and DOC (Figure 3). All the lakes with high pH showed large differences between lake types, and the second type of lake had high Chl *a* that was significantly higher than the first, third, and fifth lake types (Figure 3).

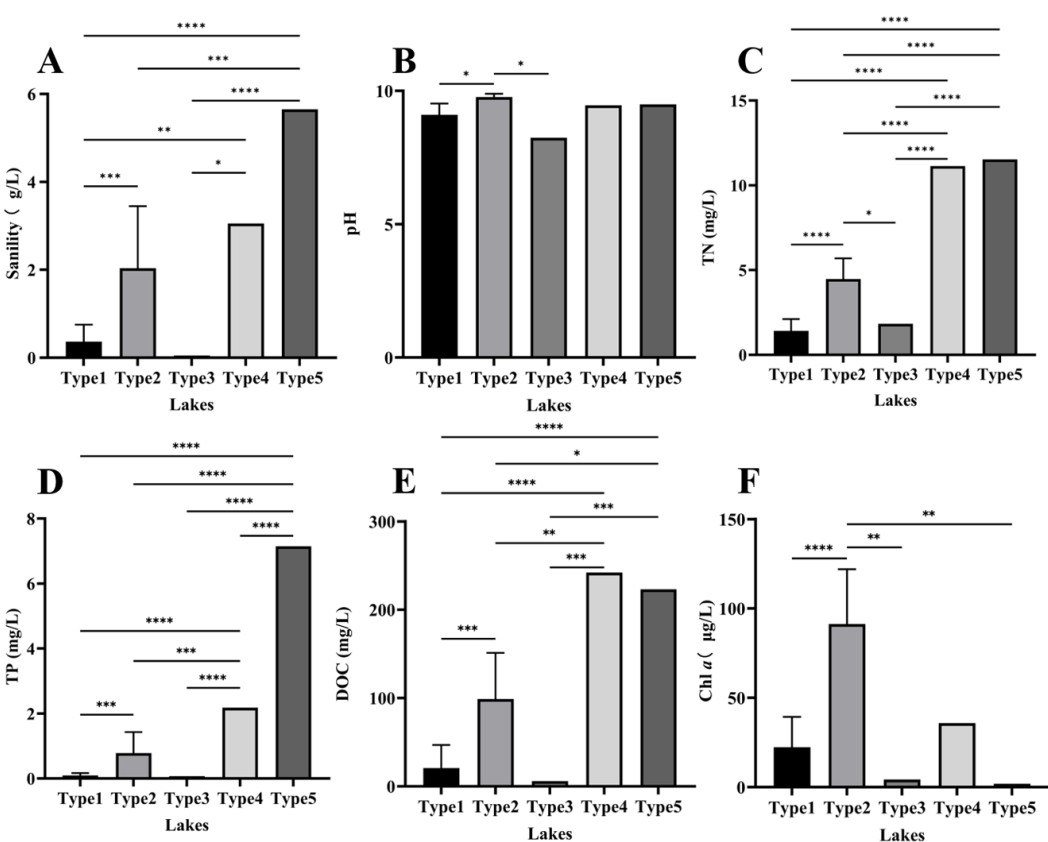

**Figure 3.** The results of an ordinary one-way ANOVA of five types of lakes in (**A**) salinity, (**B**) pH, (**C**) TN, (**D**) TP, (**E**) DOC, and (**F**) Chl *a*. Significant effects are marked by asterisks ($p < 0.0001$ (****), $p < 0.001$ (***), $p < 0.01$ (**), and $p < 0.05$ (*)).

### 3.2. Calanoid Copepod Taxonomy and Distribution

Ten calanoid species were found in 31 lakes, belonging to three families and five genera (Table 1). The most frequently occurring species were *Sinocalanus doerrii* (Brehm, 1909), *Boeckella triarticulata* (Thomson G.M., 1883), and *Sinodiaptomus sarsi* (Rylov, 1923), with frequencies of 39.4%, 21.2%, and 18.2%, respectively. *Arctodiaptomus rectispinosus* Kikuchi K., 1940, and *Neutrodiaptomus genogibbosus* Shen, 1956, occurred in three lakes. *Neutrodiaptomus pachypoditus* (Rylov, 1925), *Heterocope soldatovi* Rylov, 1922, *Acanthodiaptomus pacificus* (Burckhardt, 1913), *Metadiaptomus asiaticus* (Uljanin, 1875), and *Sinodiaptomus chaffanjoni* (Richard, 1897) occurred in only one lake. The average biomass of calanoids in the five types was 0.326 mg/L, 0.300 mg/L, 0.366 mg/L, 0.318 mg/L, and 0.546 mg/L, respectively, and there was no significant difference ($p > 0.05$) in the average biomass of calanoids between the different types. The highest biomass of calanoid copepods was 2.281 mg/L in Lake Dichi (including copepodites), while the lowest was 0.004 mg/L in Lake Datun. The density of calanoids ranged from 0.13 ind./L (Lake Sayong) to 120.83 ind./L (Lake Liming), and there was no significant difference ($p > 0.05$) in the abundance of calanoids between the different types.

The distribution of calanoids varied markedly among the sampling lakes (Figure 2). In our study, only one species, *H. soldatovi*, occurred in fresh Lake Dalijia, which is a new record for the fauna of China. The main characteristics of this species are summarised below. Female: body length (exclusive of caudal setae) 1.32–1.56 mm (n = 5). The fourth

and fifth pedigers fused, pediger V bearing two round blunt posterior corners without any spine or seta (Figure 4A). Genital plate with two large, pointed, curved processes on each side, central with five evenly short, tooth-like, blunt protrusions. Genital somite with 4–5 rows of small spines (Figure 4B). P1–P5 symmetrical, P5bsp1 wide, P5bsp2 inner convex, with a small, conical spine on the lateral posterior corner, and a line of hairs along the inner margin. P5exp1 elongated, with a small lateral spine and a row of medial hairs. P5exp2 with two short lateral spines and four slender, denticulate processes on the medial margin, of which follows a long, posterior-curved, terminal-serrated, acute process (Figure 4C). Male: body length (exclusive of caudal setae) 1.24–1.5 mm (n = 3). The body is more elongated than the female (Figure 4D). P5 uniramous and asymmetrical with wide coxopodite. A small bulge or big protuberance on the inner middle of right P5bsp2 and a small seta along the outer margin. Right P5exp single-segmented and round blunt without any seta or spine. A long, curved, U-shaped process from the left P5bsp2, extending inside the rear, with a smooth surface. The left P5exp1 rectangular and relatively short, with a small spine at the inner margin. Left P5exp2 elongated with three short spines along the outer margin which are at about one-third, two-thirds, and the end of the outer margin, respectively, and a long, straight, strong spine extends from the middle end of exp2. A row of fine hairs in the middle of the inner margin of left P5exp2, from two-thirds of the rear surface to the top, with 8–9 conical upward convex spines (Figure 4E, F). The right A1 consists of 23 segments, with the expansion on segments 18th–20th (Figure 4G).

**Table 1.** Checklist, distribution, and occurrence frequency of calanoid copepods in North-eastern (NE) China.

| Species | Distribution | Type of Habitats | Frequency (%) |
|---|---|---|---|
| Family Temoridae Giesbrecht, 1893 | | | |
| *Heterocope soldatovi* Rylov, 1922 | Dalijia | 1 | 3.0 |
| Family Centropagidae Giesbrecht, 1893 | | | |
| *Boeckella triarticulata* (Thomson G.M., 1883) | Xingkai, Sanjiaolong, Yuejin, Xiaolonghu, Qianzi, Hala, Hulun | 1, 2 | 21.2 |
| *Sinocalanus doerrii* (Brehm, 1909) | Xingkai, Dalijia, Xiaoxingkai, Delong, Jingpo, Yuebing, Wanghua, Xihulu, Sanyong, Liming, Chagan, Yangsha, Qianliujia, Datun | 1 | 39.4 |
| Family Diaptomidae Baird, 1850 | | | |
| *Acanthodiaptomus pacificus* (Burckhardt, 1913) | Dalong | 1 | 2.8 |
| *Arctodiaptomus rectispinosus* Kikuchi K., 1940 | Xiuyixi, Nvzi, Zhenzi | 2, 4 | 9.1 |
| *Neutrodiaptomus genogibbosus* Shen, 1956 | Dujuan, Luming, Dichi | 1 | 9.1 |
| *Neutrodiaptomus pachypoditus* (Rylov, 1925) | Jingpo | 3 | 3.0 |
| *Metadiaptomus asiaticus* (Uljanin, 1875) | Woniu | 5 | 3.0 |
| *Sinodiaptomus chaffanjoni* (Richard, 1897) | Hama | 2 | 3.0 |
| *Sinodiaptomus sarsi* (Rylov, 1923) | Qingken, Wanghua, Qianliujia, Wangzi | 1, 2 | 18.2 |

* Type 1: plain and mountain barrier lakes, mostly eutrophic freshwater lakes; Type 2: hyposaline shallow lakes with high organic matter (except Lake Qianzi); Type 3: Lake Jingpo, mountain barrier lake; Type 4: Lake Xiuyixi, eutrophic hyposaline lake; Type 5: Lake Woniu, shallow with high mineralisation lake.

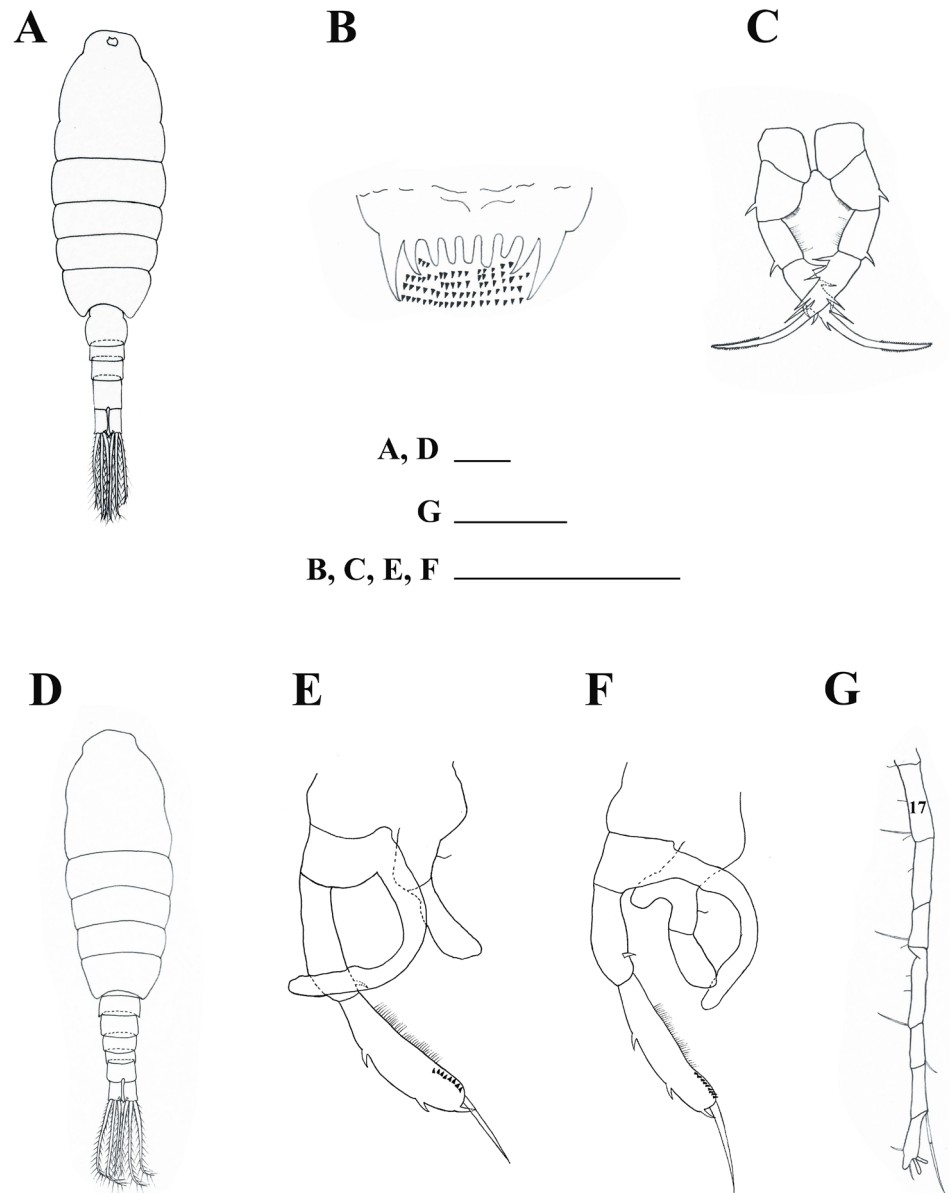

**Figure 4.** *Heterocope soldatovi* Rylov, 1922 from Lake Dalijia, female (**A–C**) and male (**D–G**). (**A**) Female habitus; (**B**) genital plate; (**C**) the fifth leg; (**D**) male habitus; (**E,F**) the fifth leg; (**G**) right antennule (segments 17th–23rd). Scale bar: 0.2 mm.

Of the two species of Centropagidae, *S. doerrii* was widely distributed in all types of lakes and was mostly the dominant species in the lakes. It was found in 14 lakes located in plain lakes with an average altitude of 129 m a.s.l., salinity range of 0.15–3.34 g/L, and Chl *a* range of 9.97–42.02 µg/L. Two other species also occurred in three lake types, *B. triarticulata* and *S. sarsi*. *B. triarticulata* was a sole calanoid species in 86% of the lakes where it occurred. It occurred at both high and low altitudes (ranging from 55 m a.s.l. to 833 m a.s.l.), and its distribution was patchy and wide.

*S. sarsi* was common in fresh and subsaline lakes with a salinity range of 0.45–3.34 g/L. In addition, its habitat usually had high DOC. *S. chaffanjoni* was only found in Lake Hama, with a low biomass (0.09 mg/L). The distribution of *N. genogibbosus*, *N. pachypoditus*, and *A. pacificus* was restricted to volcanic lakes. Furthermore, *N. genogibbosus* inhabited high-altitude (>1100 m a.s.l.) lakes (Lake Dujuan, Luming, and Dichi). *N. pachypoditus* was only found in Lake Jingpo (mountain barrier lake). In the case of *A. rectispinosus* and *M.*

*asiaticus*, they were found in both saline and alkaline lakes, such as Lake Xiuyixi, Nvzi, Zhenzi, and Woniu (salinity range from 2.86 to 5.65 mg/L).

In addition, two calanoid species coexisted and were found in five lakes. All of them contained *S. doerrii*, and these five water bodies are type 1 and type 3, including both freshwater and subsaline lakes. The t-test analysis showed that there were significant differences in body length between the two coexisting species in Lakes Xingkai and Lake Wanghua and in biomass in Lakes Jingpo and Lake Dalijia (Table 2, Table S2).

**Table 2.** T-test results of body length and biomass of five lakes with two calanoid species coexisting.

| Lake | Species | Body Length | | Biomass | |
|---|---|---|---|---|---|
| | | *t* | *p* | *t* | *p* |
| Lake XK | *S. doerrii–B. triarticulata* | -3.803 | 0.019 | 0.899 | 0.419 |
| Lake JP | *S. doerrii–N. pachypoditus* | -2.527 | 0.065 | 3.259 | 0.031 |
| Lake WH | *S. doerrii–S. sarsi* | -2.881 | 0.045 | 2.528 | 0.065 |
| Lake QLJ | *S. doerrii–S. sarsi* | 0.062 | 0.954 | -0.025 | 0.982 |
| Lake DLJ | *S. doerrii–H. soldatovi* | -0.356 | 0.740 | 4.293 | 0.013 |

In our study, cyclopoids occurred with calanoids in 29 lakes, mainly including *Mesocyclops dissimilis* Defaye and Kawabata, 1993, *Thermocyclops taihokuensis* Harada, 1931, and *Thermocyclops kawamurai* Kikuchi, 1940. The biomass ratio of calanoids to cyclopoids varied considerably from lake to lake (Figure 5), ranging from 0.02 (Lake Sangyong) to 37.04 mg/L (Lake Dichi). However, no significant relationship between the biomass ratio and environmental variables was observed.

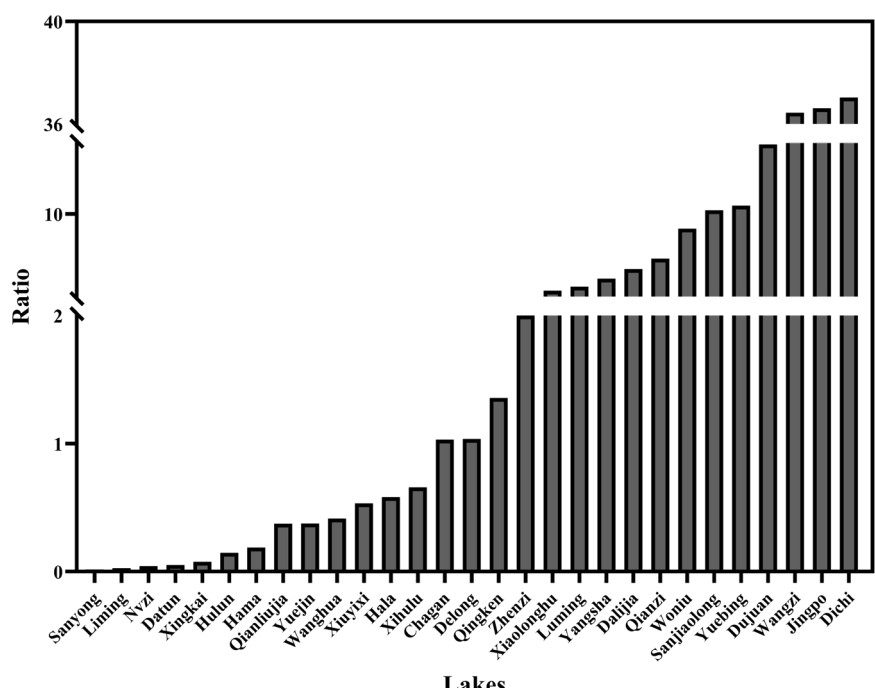

**Figure 5.** The biomass ratio of Calanoida to Cyclopoida in lakes.

*3.3. Environmental Variables and Correlation Analysis*

For type 1 lakes, the first two axes of the RDA ordination explained 44.87% of the variance (Figure 6A). The eigenvalues for axis 1 and axis 2 were 0.257 and 0.146, respectively. Calanoid copepods were significantly correlated with altitude (pseudo-*F* =12.9, *p* = 0.002), NO$_3$-N (pseudo-*F* =9.2, *p* = 0.004), DOC (pseudo-*F* =5.3, *p* = 0.006), and TP (pseudo-

$F$ =4.2, $p$ = 0.016) (Figure 6A). For type 2 lakes, the first two axes of the CCA ordination explained 88% of the variance (Figure 6B). Within these variables, conductivity (pseudo-$F$ =6.6, $p$ = 0.002), TN (pseudo-$F$ = 9.7, $p$ = 0.002), and TP (pseudo-$F$ =11.3, $p$ = 0.006) were all significantly correlated with calanoid copepods. Both CCA and RDA indicated that TN and TP were important environmental factors affecting the distribution and diversity of calanoid species in NE China.

The CCA and RDA results also reflected the differences in the habits of individual species and showed that nutrients played an important role in the growth of these species. In type 1 lakes, *S. doerrii* was negatively correlated with altitude, TP, and DOC but positively correlated with NO₃-N. *N. genogibbosus* was significantly positively correlated with altitude, indicating a preference for high-altitude waters. *S. sarsi* was significantly positively correlated to TP and DOC and negatively correlated with NO₃-N. The environmental factors affecting *B. triarticulata* varied according to the type of lake. In type 1 lakes, it was positively correlated with DOC and TP and negatively correlated with NO₃-N. In type 2 lakes, however, the species was negatively correlated with TP and conductivity. In type 2 lakes, *S. chaffanjoni* was significantly positively correlated with TN and negative with conductivity, and TP. *A. rectispinosus* was significantly positively correlated with TP and conductivity, negative with TN (Figure 6).

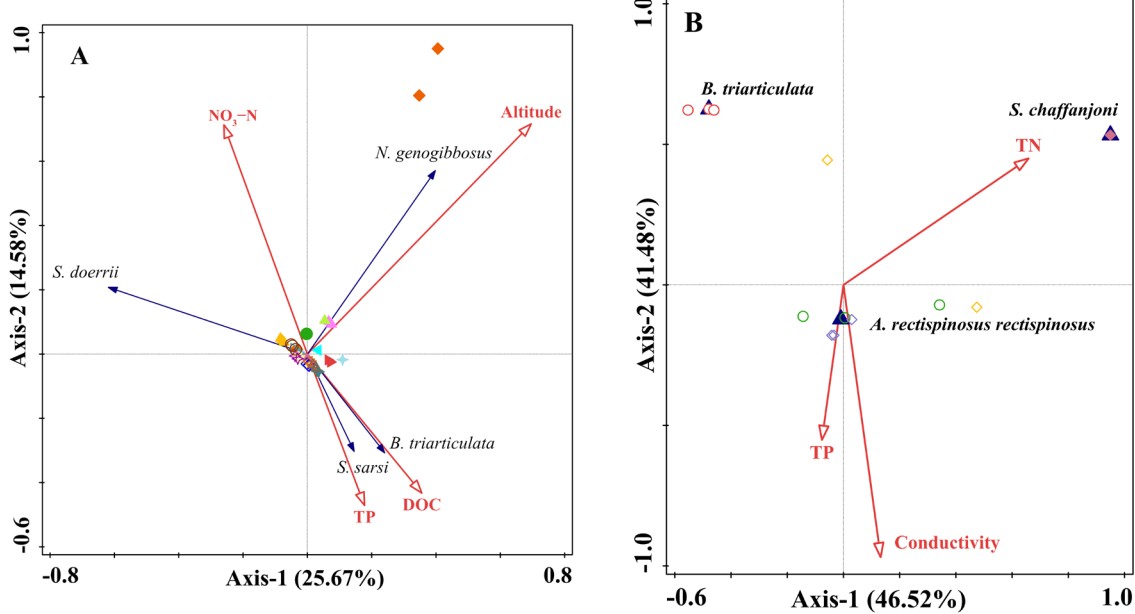

**Figure 6.** Relationship between calanoid copepods and environmental factors, respectively, in (**A**) type 1 lakes and (**B**) type 2 lakes.

## 4. Discussion

A total of 37 lakes were surveyed, and the presence of calanoid copepods was recorded in 31 lakes. The diversity of calanoid copepods is high in NE China. To date, 21 calanoid species have been found in this area (Table 3). The ten species belong to eight genera, and three families were found in the present study, in which the species richness of the family Diaptomidae was the highest. We found that the species of the family Centropagidae had a wider distribution than the other two families, indicating that they had a greater adaptability and dispersal ability. Compared with eutrophic lakes in South-western China and the Yangtze River Basin region, calanoids exhibited higher abundance and biomass in lakes in NE China [42,43].

Family Temoridae Giesbrecht, 1893

Within the family Temoridae, there are three genera distributed in inland waters, of which *Epischura* and *Heterocope* have been found in NE China. *E. chankensis* and *H. appendiculata* were recorded in Lake Xinkai and Lake Jingpo by Shen and Song (1979) [7] but were not found in our study. Another species, *H. soldatovi*, was found for the first time in China. The genus *Heterocope* is mostly distributed in fresh or brackish water in the Palaearctic realm [34]. *H. soldatovi* was widely distributed in the lower reaches of the Amur River (called the Heilongjiang River in China) in Russia [34]. Compared with the description from Russia, the male character is variable and unstable in morphology; the bsp2 of the right P5 was short and broad, with a small bulge in the middle of the inner margin in the Amur River's population [44] but without a bulge in the Lake Bolon population [45]. Our specimens have a small bulge or large protuberance in the right P5bsp2 and a blunt end of the right P5exp; both characteristics are similar to the Amur River's population.

Family Diaptomidae Baird, 1850

At present, there are 58 species of Diaptomidae in China [6,46,47]. According to climatic zones, there are 9, 21, 33, and 9 species of Diaptomidae in the Qinghai–Tibet Plateau, temperate, subtropical, and tropical zones of China, respectively. A total of 15 species of the family Diaptomidae were found in the NE area, accounting for 71.4% and 26.3% of Diaptomidae in the temperate zone and all of China, respectively. It indicated a rich biodiversity of Diaptomidae in the NE area. Its species richness is higher than 5 species in South Korea and 11 species in Japan [33,48].

*Metadiaptomus asiaticus* is found only in Asia and Russia, including northern China, Mongolia, Turkey, Iran, Kazakhstan, and Russia [49,50]. It prefers to live in highly chlorinated and alkaline waters [51]. However, this species can also adapt to highly eutrophicated waters (TN-11.53 mg/L, TP-7.15 mg/L), such as Lake Woniu in this study.

*Acanthodiaptomus pacificus* is a rare cryophilic species and superior competitor in high-altitude ponds and lakes [7,33]. It is also found in Korea, Japan, and Russia. In our study, *A. pacificus* was found in Lake Dalong, an oligotrophic deep crater lake.

There are ten species/subspecies of *Arctodiaptomus* in northern and western China. Three species *A. rectispinosus*, *A. stewartianus*, and *A. wierzejskii*, are distributed in the NE area. In the present study, we only found *A. rectispinosus*, which is widely distributed in the saline alkaline waters of NE China. The lakes (Lake Nvzi, Zhenzi, and Xiuyixi) inhabited by *A. rectispinosus* are subsaline and hypereutrophic, indicating its high adaptability.

Like *Arctodiaptomus*, *Neutrodiaptomus* is the most diverse genus of species in China. *N. incongruens*, *N. pachypoditus*, and *N. genogibbosus* are distributed in the NE area, and the latter two were found in the present study. *N. pachypoditus* is a cryophilic species found in lakes, rivers, and ponds at high latitudes (above 40° N) and is also distributed in Japan and Russia [33,52]. In our study, *N. pachypoditus* was found in Lake Jingpo, a deep volcanic barrier lake with low primary production. *N. genogibbosus* was only widely distributed in lakes, ponds, and rivers in NE China [7]. In our study, this species was found in three volcanic lakes (Dichi, Dujuan, and Luming) with an altitude of more than 1100 m a.s.l. and low primary production.

Four *Sinodiaptomus* species have been reported from China, of which *S. sarsi* and *S. chaffanjoni* are distributed in the NE area. *S. sarsi* is widely distributed in Eastern, Southern, and Central Asia and Eastern Europe, with some areas showing dispersal [7,53]. This species prefers the small, shallow, high DOC lakes of the NE area as recorded in a previous study [7]. *S. chaffanjoni* is distributed only in NE China and was found in Lake Hama, which is subsaline with high conductivity, nitrogen, and DOC.

Family Centropagidae Giesbrecht, 1893

Non-marine centropagids are distributed in Australasia, sub-Antarctic Island, the Antarctic Peninsula, nearby islands, and southern and high-altitude portions of South America [54]. Centropagids were thought to be derived from marine ancestors, with transitions running from marine to estuarine to freshwater to inland saline. Nowadays, the

fresh waters of the Southern Hemisphere and the Northern Hemisphere are colonised by Centropagidae and Diaptomidae, respectively [55]. However, the species of the family Centropagidae had a greater distribution in our study, perhaps due to their evolutionary euryhalinity and ability to produce resting eggs [56] and the continuous morphological reduction in the setation and segmentation of the swimming legs accompanied by the ecological sequence [57]. Another reason may be that centropagids are particularly adapted to colonise newly available and extreme environments, such as lakes formed at the margins of ice sheets, as in our study [56]. Therefore, the high latitude of our sampling area and the long winter freeze period of the lakes may be suitable for their growth. In China, two genera (*Boeckella* and *Sinocalanus*) of centropagids were found in inland waters, both of which were distributed in NE lakes. Both the *Boeckella* group and *Sinocalanus* are considered to represent independent freshwater colonisations [58].

At present, about 53 species of the genus *Boeckella* have been discovered worldwide, mainly distributed in the Australasian and Neotropical regions [59]. *Boeckella triarticulata* is mainly distributed in Australia and New Zealand but is also found in China, Mongolia, Russia [4,54], and Italy [60]. In China, it was previously recorded as *B. orientalis* [7]; Guo (1999, unpublished data) suggested that *B. orientalis* should be a subspecies of *B. triarticulata* based on their morphological differences. In the absence of a molecular sequence comparison, the status of this species needs to be verified. Several studies suggest that waterbirds may play an important role in the long-distance dispersal of ephippia, statoblasts, and other propagules [61–63]. Therefore, a possible reason for the dispersal of *B. triarticulata* to NE China would be related to the bird migration route along eastern Australia [64]. This route includes the eastern and north-eastern parts of China. If this is the case, it is evident that the environment in NE China is suitable for the colonisation of this species, whereas it is not in the eastern part, possibly due to the high water temperature. In our study, *B. triarticulata* was found in seven lakes adjacent to the distribution range of *S. doerrii*. These lakes are either freshwater or brackish, consistent with its characteristics as a salt-tolerant species [57], with trophic levels ranging from oligotrophic to hypereutrophic, indicating a strong adaptability of *B. triarticulata*.

The genus *Sinocalanus* comprises five species that occur in estuarine and freshwater environments and may have diversified into freshwater at present [58]. Among them, two species are completely adapted to the freshwater environment, *S. tenellus* and *S. doerrii*, both of which are distributed in NE China [65,66]. We only found *S. doerrii* in 14 freshwater lakes, accounting for 45% of the lakes surveyed. *S. doerrii* is widely distributed in the subtropical and temperate zones of China (Shen and Song, 1979) [7], with no records in the NE area until a decade ago [65]. Previous studies indicated that it has extensive adaptability, competitive advantage, and salinity adaptation; the above characteristics make it a potential invasive species [18]. The invasion pathway may be related to the introduction of two filter-feeding fishes (*Hypophthalmichthys molitrix* and *H. nobilis*) from the subtropical lakes of China [67]. The introduction of organisms such as fish can lead to species invasions in NE China, affecting the aquatic ecosystem. Therefore, human activities should focus on or avoid biological invasions.

In NE lakes, only *S. doerrii* was found to coexist with other calanoids in five lakes, including *B. triarticulata*, *N. pachypoditus*, *S. sarsi*, and *H. soldatovi*. We also found that the species (*S. doerrii* and *B. triarticulata*) of the family Centropagidae had a wider distribution than the other two families. Within the world's inland waters, Diaptomidae and Centropagidae generally have mutually exclusive distributions, although some exceptions are known [54]. In our study, except for Lake Dalijia (coexisting with Temoridae), species of Diaptomidae and Centropagidae coexisted in the other four lakes. Furthermore, *S. doerrii* was dominant in all five lakes, indicating that a large-scale invasion by this species is quite possible and consistent with the strong dispersal seen in the western United States [18]. Hutchinson (1967) [68] asserted that in freshwater calanoid copepods, the co-occurrence of different but similar sized species is very rare. The body length of the two calanoid species was significantly different in Lake Xingkai and Lake Wanghua but similar in Lake

Jingpo and Lake Qianliujia. Although there was no significant difference in the biomass of the two species in Lake Xingkai, Wanghua, and Qianliujia, *S. doerrii* still showed a high dispersal capacity. In Lake Jingpo, *S. doerrii* could breed prosperously in various environments and under different predation pressures, potentially outcompeting the Diaptomidae family [69]. In Lake Dalijia, another reason for the co-occurrence of the two species may be that *S. doerrii* was mostly found in lakes and rivers, which was similar to the habitat of *H. soldatovi*. These two filter-feeding species may occupy different ecological niches to coexist. Additionally, studies have shown that under conditions of food shortage and high adult densities, copepod populations can even be maintained by older stages feeding on specific early life stages [70,71]. We found that most *H. soldatovi* individuals belonged to copepodites, possibly due to predation on nauplii by themselves or other copepods in this lake.

The Relationship between Calanoid Species and environmental variables

In temperate shallow lakes, temperature and nutrients are the most important environmental factors [72]. The RDA results showed that altitude played an important role in the growth and distribution of most calanoid species in NE China. With increasing altitude, the number of calanoid species gradually decreased. There were only three and one calanoid species in lakes above 500 m and 1000 m a.s.l., respectively, and these species are all stenothermic. This result is similar to the small number of calanoid species found at high altitudes in Japan [33]. Nutrients can affect the growth and development of phytoplankton, which in turn indirectly affects the species and abundance of zooplankton through the transfer of food chains [73,74]. Our result showed that in both types of lakes, the factor with the strongest correlation was TP. In the central area of our study regions, the majority of lakes are small saline alkaline lakes. Due to the poor soils, low resources, and low zooplankton in small saline and alkaline lakes, the lakes are fertilised to improve the environment for fish growth, improve water quality, and develop fish farming. Such high nitrogen and calcium superphosphate fertilisation changes the nutrient concentration of the lake and significantly increases the nitrogen and phosphorus content of the lake. In summer, the monsoon climate in this area allows for large amounts of precipitation, which leads to the growth and reproduction of large zooplankton, such as calanoid species *Daphnia* in the lake. This may be one of the reasons why the results of the RDA and CCA analyses showed that the distribution of calanoid species was strongly correlated with TN and TP. The degree of water mineralisation was one of the main factors determining the specific characteristics of the species composition and the number of zooplankton species [75]. In this study, with the increase in mineralisation, the calanoid species in the lake gradually became single, and only three species were distributed in hyposaline lakes. Our results showed that *A. rectispinosus* was positively related to conductivity, consistent with previous records that this species was a halophilic species [76]. Thus, the diversity of calanoid copepods in NE China is high, while eutrophication and mineralisation are affecting its survival and diversity. In the future, attention should be paid to environmental protection in order to maintain the diversity of mineralisation in NE China.

**Table 3.** Updated list of calanoid copepods recorded in NE China.

| | Species | This Study | Shen and Song (1979) | Yu et al. (2001–2017) | Distribution |
|---|---|---|---|---|---|
| | Family Temoridae Giesbrecht, 1893 | | | | |
| 1 | *Epischura chankensis* Rylov, 1928 | | √ | √ | China and Russia [77]. |
| 2 | *Heterocope appendiculata* Sars G.O., 1863 | | √ | | China and Russia [34], Ukraine [78], Turkey [79], Germany [80], Iceland [81], |

| # | Species | | | | Distribution |
|---|---|---|---|---|---|
| | | | | | Finland [82], Sweden [83], and Poland [84]. |
| 3 | *Heterocope soldatovi* Rylov, 1922 | √ | | | China and Far-Eastern Russia [45]. |
| | Family Centropagidae Giesbrecht, 1893 | | | | |
| 4 | *Boeckella triarticulata* (Thomson G.M., 1883) | √ | √ | | China, New Zealand [85], Australia [27], Eastern Mongolia, Far-Eastern Russia, and Italy [4,69]. |
| 5 | *Sinocalanus doerrii* (Brehm, 1909) | √ | | √ | China and the United States [86]. |
| 6 | *Sinocalanus tenellus* (Kikuchi, 1928) | | | √ | China [66], Japan [87], and South Korea [88]. |
| | Family Diaptomidae Baird, 1850 | | | | |
| 7 | *Arctodiaptomus wierzejskii* (Richard, 1888) | | √ | √ | China, Mongolia [89], Russia [45], Tunisia [90], and France [91]. |
| 8 | *Acanthodiaptomus pacificus* (Burckhardt, 1913) | √ | √ | √ | China, Japan [33], and South Korea [92]. |
| 9 | *Arctodiaptomus rectispinosus* Kikuchi K., 1940 | √ | | | China. |
| 10 | *Arctodiaptomus stewartianus* (Brehm, 1925) | | | √ | China and India [93]. |
| 11 | *Eudiaptomus graciloides* (Lilljeborg, 1888) | | | √ | China and European lakes [94]. |
| 12 | *Mongolodiaptomus birulai* (Rylov, 1922) | | √ | √ | China, Vietnam, and the Philippines [95]. |
| 13 | *Heliodiaptomus kikuchii* Kiefer, 1932 | | √ | | China, the Korean Peninsula, Japan, and Indonesia [33,96]. |
| 14 | *Neodiaptomus schmackeri* (Poppe and Richard, 1892) | | | | China, Japan [33], South Korea [92], Kazakhstan [50], India [97], and Thailand [98]. |
| 15 | *Neutrodiaptomus genogibbosus* Shen, 1956 | √ | √ | | China and Kazakhstan [50]. |
| 16 | *Neutrodiaptomus incongruens* (Poppe, 1888) | | √ | | China, Russia [52], and Kazakhstan [50]. |
| 17 | *Neutrodiaptomus pachypoditus* (Rylov, 1925) | √ | √ | √ | China, Japan [33], and Russia [52]. |
| 18 | *Metadiaptomus asiaticus* (Uljanin, 1875) | √ | | | China, Mongolia [51], and Russia [50,99]. |
| 19 | *Sinodiaptomus chaffanjoni* (Richard, 1897) | √ | √ | √ | China [7,32]. |
| 20 | *Sinodiaptomus sarsi* (Rylov, 1923) | √ | √ | | Eastern Asia [6,33,92], Turkey [100], Romania [101], Kazakhstan [50], and Ukraine [102]. |
| 21 | *Tropodiaptomus oryzanus* Kiefer, 1937 | | √ | | China [32], South Korea [92], Vietnam, Cambodia, and Thailand [3,103]. |

## 5. Conclusions

The richness of calanoid copepods is high in NE China and may increase with the number of lakes surveyed. In our study, ten calanoid species belong to eight genera, and

three families were found in NE China. Species of the family Centropagidae had a greater distribution, especially *Sinocalanus doerrii* and *Boeckella triarticulata*. To date, 21 calanoid species are prospective to be distributed in NE China. Our results showed that environmental factors influencing the distribution of calanoid species were different in various types of lakes. We hope that our research will contribute to a further understanding of the diversity and habitat of calanoid copepods in China and Asia.

**Supplementary Materials:** The following supporting information can be downloaded at https://www.mdpi.com/article/10.3390/d16050288/s1, Figure S1: The biomass (**A**) and density (**B**) of calanoid copepods in 31 lakes; Table S1: Physical and chemical characteristics of 31 lakes with calanoid copepod occurrences; Table S2: The body length (average ± standard deviation) of calanoid copepods (adult) in 31 lakes.

**Author Contributions:** Conceptualisation, R.D. and F.C.; methodology, R.D.; software, R.D.; formal analysis, R.D., Y.L., and L.L.; investigation, L.L. and F.C.; resources, R.D., F.C., and S.S.; data curation, R.D.; writing—original draft preparation, R.D.; writing—review and editing, F.C., S.S., Y.L., and L.L.; funding acquisition, F.C. All authors have read and agreed to the published version of the manuscript.

**Funding:** This research was funded by the National Natural Science Foundation of China (No. 32271637, 32371631).

**Data Availability Statement:** Data are contained within the article.

**Acknowledgments:** We would like to thank Yuan Li for assistance with sample collection and anonymous referees for commenting on the manuscript.

**Conflicts of Interest:** The authors declare no conflict of interest.

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
