# Peer review of "Diversity of Freshwater Calanoid Copepods (Crustacea: Copepoda: Calanoida) in North-Eastern China"

_diversity, doi:10.3390/d16050288_

Round 1

Reviewer 1 Report

Comments and Suggestions for Authors

The authors investigated the species diversity of freshwater calanoid copepods, and the environmental factors assorted with their occurrences in North-eastern (NE) China. The sampling methods and the statistical analyses employed seem to be sound. Overall, the manuscript is interesting. The results are useful in understanding the biogeographical distribution of freshwater calanoid copepods in NE China in the global context.

Comments on the Quality of English Language

My comments are primarily of editorial concern. The English grammar should be checked again throughout the text.

Comments:

Title and other places: “Northeastern” to be “North-eastern”?

Line 33: Please change “are” to “is composed of” or “consists of”, noting that Diaptomidae ‘is’ a family of freshwater copepods.

Lines 66 to 75: I presume the authors describe the current/present conditions of the study area, that are the same as or very similar to those when the survey was undertaken in 2019. If so, the present tense should be used in the sentences.  For example, “The Siberian continental air mass is controlled in winter by…” instead of “The Siberian continental air mass was controlled in winter by…”

Line 95: “samples preserved in 4% formaldehyde solution” to be “and samples were preserved in 4% formaldehyde solution”

Line 158: “The average biomass of the five types” to be “The average biomass of calanoids in the five lake types”

Line 160: “there was no significant difference (P >0.05) between the different types” to be “there was no significant difference (P >0.05) in the average biomass of calanoids between the different lake types”

Lines 193-194: “Two species of Centropagidae, S. doerrii were widely distributed in all types of lakes and were mostly the dominant species in the lakes” to be “Of the two species of Centropagidae, S. doerrii was widely distributed in all types of lakes and was mostly the dominant species in the lakes”

Lines 197-198: “Most B. triarticulata were sole calanoid copepods and were present in 86% of the lakes where they occurred.” – The meaning of this sentence is unclear. It needs to be re-written. Do you mean “Boeckella triarticulata was a sole calanoid species in 86% of the lakes where this species occurred”?

Line 201: “its habitat is” to be “its habitat was”

Lines 204-205: “inhabited high-altitude lakes (Lake Dujuan, Luming, and Dichi), these three lakes were all above 1100 m a.s.l.” to be “inhabited high-altitude (>1100 m a.s.l.) lakes (Lake Dujuan, Luming, and Dichi)”

Lines 209-210: Please delete “It is very interesting that” and start this sentence as “All of them…”.

Line 220: “ratio” to be “biomass ratio”

Line 238: What does “It” indicate?

Line 257: “is” to be “was”

Author Response

Dear editors and reviewer,

We feel great thanks for your professional review work on our article. As you are concerned, there are several problems that need to be addressed. According to your nice suggestions, we have made extensive corrections to our previous draft. For your suggestions, we have highlighted the corrections in the manuscript in green. The detailed corrections are listed below.

Title and other places: “Northeastern” to be “North-eastern”?

Reply: We have corrected “Northeastern” to “North-eastern” in title and other places (Lines 14, 37, 200).

Line 33: Please change “are” to “is composed of” or “consists of”, noting that Diaptomidae ‘is’ a family of freshwater copepods.

Reply: We have corrected “are” to “consists of” (Line 35).

Lines 66 to 75: I presume the authors describe the current/present conditions of the study area, that are the same as or very similar to those when the survey was undertaken in 2019. If so, the present tense should be used in the sentences.  For example, “The Siberian continental air mass is controlled in winter by…” instead of “The Siberian continental air mass was controlled in winter by…”

Reply: We have corrected “The Siberian continental air mass was controlled in winter by a long glacial period” to “The Siberian continental air mass is controlled in winter by a long glacial period” (Line 68).

Line 95: “samples preserved in 4% formaldehyde solution” to be “and samples were preserved in 4% formaldehyde solution”

Reply: We have corrected “samples preserved in 4% formaldehyde solution” to “and samples were preserved in 4% formaldehyde solution” (Line 97).

Line 158: “The average biomass of the five types” to be “The average biomass of calanoids in the five lake types”

Reply: We have corrected “The average biomass of the five types” to “The average biomass of calanoids in the five lake types” (Lines 168-169).

Line 160: “there was no significant difference (P >0.05) between the different types” to be “there was no significant difference (P >0.05) in the average biomass of calanoids between the different lake types”

Reply: We have corrected “there was no significant difference (P >0.05) between the different types” to “there was no significant difference (P >0.05) in the average biomass of calanoids between the different lake types” (Line 170).

Lines 193-194: “Two species of Centropagidae, S. doerrii were widely distributed in all types of lakes and were mostly the dominant species in the lakes” to be “Of the two species of Centropagidae, S. doerrii was widely distributed in all types of lakes and was mostly the dominant species in the lakes”

Reply: We have corrected “Two species of Centropagidae, S. doerrii were widely distributed in all types of lakes and were mostly the dominant species in the lakes” to “Of the two species of Centropagidae, S. doerrii was widely distributed in all types of lakes and was mostly the dominant species in the lakes” (Lines 206-207).

Lines 197-198: “Most B. triarticulata were sole calanoid copepods and were present in 86% of the lakes where they occurred.” – The meaning of this sentence is unclear. It needs to be re-written. Do you mean “Boeckella triarticulata was a sole calanoid species in 86% of the lakes where this species occurred”?

Reply: We have rewritten this sentence to “B. triarticulata was a sole calanoid species in 86% of the lakes where it occurred” (Line 210).

Line 201: “its habitat is” to be “its habitat was”

Reply: We have corrected “its habitat is” to “its habitat was” (Line 214).

Lines 204-205: “inhabited high-altitude lakes (Lake Dujuan, Luming, and Dichi), these three lakes were all above 1100 m a.s.l.” to be “inhabited high-altitude (>1100 m a.s.l.) lakes (Lake Dujuan, Luming, and Dichi)”

Reply: We have corrected “inhabited high-altitude lakes (Lake Dujuan, Luming, and Dichi), these three lakes were all above 1100 m a.s.l.” to “inhabited high-altitude (>1100 m a.s.l.) lakes (Lake Dujuan, Luming, and Dichi)” (Lines 216-217).

Lines 209-210: Please delete “It is very interesting that” and start this sentence as “All of them…”.

Reply: We have rewritten this sentence to “All of them contained S. doerrii and these five water bodies are type 1 and type 3, including both freshwater and sub-saline lakes” (Lines 221-223).

Line 220: “ratio” to be “biomass ratio”

Reply: We have corrected “ratio” to “biomass ratio” (Line 231).

Line 238: What does “It” indicate?

Reply: It indicated the result of CCA and RDA. There was a problem with our formulation, so the sentence was rewritten to “CCA and RDA results also reflected the differences in the habits of individual species and shows that nutrients played an important role in the growth of these species” (Lines 250-251).

Line 257: “is” to be “was”

Reply: We have corrected “is” to “was” (Line 270).

Reviewer 2 Report

Comments and Suggestions for Authors

This is a well-written manuscript on calanoid copepod diversity in Northeast China's little-known remote area. In general, the contents are acceptable, but there are some typing errors and mistakes to be corrected; please see the details in the attached file. Please keep only the necessary references from the list of 96, as some may be outdated.

Comments on the Quality of English Language

English language requires minor editing.

Author Response

Dear editors and reviewer,

We feel great thanks for your professional review work on our article. As you are concerned, there are several problems that need to be addressed. According to your nice suggestions, we have made extensive corrections to our previous draft. For your suggestions, we have highlighted the corrections in the manuscript in yellow. The detailed corrections are listed below.

Replace "widespeard" with "widespread" to correct the typing error.

Reply: We have corrected "widespeard" to "widespread" (Line 18).

“TP” should be written in full, not an abbreviation.

Reply: We have added the full form of TP (Line 21).

Rearrange the keywords in accordance with the first alphabet.

Reply: We have rearranged the keywords in accordance with the first alphabet in Line 26.

Replace “…were sampled in July, 2019 when the individual adulted.” with “… were sampled in July 2019, when the individual matured.”

Reply: We have rewritten the sentence to “Due to the long-frozen period in winter, copepod samples were sampled in July 2019, when the individual matured” (Lines 94-95).

To be better understood, please rephrase this sentence. “In the laboratory, …… measured under Carl Zeiss optical microscope.” to “In the laboratory, calanoid copepods were identified using the identification keys of Shen and Song [7] and Sheveleva et al. [34]. The animals were counted and measured under a Carl Zeiss optical microscope.”

Reply: We have rewritten the sentence to “In the laboratory, calanoid copepods were identified using the identification keys of Shen and Song [7] and Sheveleva et al., 2021 [34]. The animals were counted and measured under Carl Zeiss optical microscope” (Lines 98-100).

Replace two words in this sentence. “… (if not enough, measured all of them) to estimate the dry biomass following length-weight relations [38,39].” To “… (if not enough, measure all of them) to estimate the dry biomass following length-weight relationship [38,39].”

Reply: We have rewritten the sentence to “Body lengths of calanoid copepods were measured for 20 individuals (if not enough, measure all of them) to estimate the dry biomass following length-weight relationship [38,39]” (Lines 100-102).

You cited Table S1 in the content, but this table is not included in the manuscript???

Reply: We regretted that Table S1 was not included in the manuscript. We have zipped all the supplementaries into one file with name “manuscript-supplementary.zip” and uploaded it to the system.

You analyzed the data from 31 lakes which calanoids occurred, how about the other 6 lakes which calanoids did not occur. Perhaps you should also mention the names of these six lakes.

Reply: We have added the names of these six lakes in “No calanoid was found in six lakes in our survey, namely Lake Shenyangxi, Lake Zhongnei, Lake Yezi, Lake Zhouzi, Lake Wanghua, and Lake Hure” (Lines 132-133).

The abbreviations of some copepod characteristics in the manuscript such as P1-P5, P5bsp1, P5exp1, exp, A1, etc., should be mentioned in full once, may be in the Materials and Methods, after that you can use the abbreviation throughout.

Reply: We have added the full form of any abbreviation in Lines 103-106.

To be more specific, please add the word “occurrence” in front of“frequency” in the table caption.

Reply: We have added the word “occurrence” in front of “frequency” in the table caption (Line 200).

Please leave one space before numbers in the words: Type1, Type2, Type3, and Type4.

Reply: We have added one space before numbers in the words (Lines 202-204).

“Two species of Centropagidae, S. doerrii were widely distributed in all types of lakes and were mostly the dominant species in the lakes. It was found in 14 lakes located in plain lakes with an average altitude is 129 m a.s.l., salinity range of 0.15-3.34 g/L, Chl a range of 9.97-42.02 μg/L. Two other species also occurred in three lake types, B. triarticulata 196 and S. sarsi.” The above paragraph is confusing, please rewrite it. You mentioned two species of Centropagidae, but referred to only one.

Reply: We have rewritten the sentence to “Of the two species of Centropagidae, S. doerrii was widely distributed in all types of lakes and was mostly the dominant species in the lakes” (Lines 206-207).

Please correct the figure caption to the following: Figure 4. Heterocope soldatovi Rylov, 1922 from Lake Dalijia, female(A−C), and male (D−G). (A) female habitus; (B) genital plate; (C) fifth leg; (D) male habitus; (E, F) fifth leg; (G) right Antennule (segments 17th−23rd). Scale bar: 0.2 mm.

Reply: We have corrected the Figure 4 caption to “Figure 4. Heterocope soldatovi Rylov, 1922 from Lake Dalijia, female (A-C), and male (D-G). (A) female habitus; (B) genital plate; (C) the fifth leg; (D) male habitus; (E, F) the fifth leg; (G) right antennule (segments 17th-23rd). Scale bar:0.2 mm” (Lines 234-236).

You cited Figure 4A on line 232, but the correct one is Figure 6-A.

Reply: We have corrected the citation “Figure 4A” to “Figure 6-A” (Line 244)

Add the word “China” after “NE”.

Reply: We have added the word “China” after “NE” (Line 249).

Correct the typing error from “S. sarisi” to “S. sarsi”.

Reply: We have corrected the“S. sarisi” to “S. sarsi” (Line 254).

Table 3 is not cited in the content. Correct the Table 3 caption, by deleting one “in” from the sentence, please see below. Table 3. Update list of calanoid copepods recorded in NE China.

Reply: We have added the citation of Table 3 in Line 268, and corrected the caption to “Table 3. Update list of calanoid copepods recorded in NE China” (Line 275).

The authors incorrectly cited that Mongolodiaptomus birulai has been recorded in China, Vietnam, Thailand, and the Philippines [88], but according to [88], this species actually was found in China, Vietnam, Taiwan, and the Philippines.

Reply: We have rechecked the distribution of Mongolodiaptomus birulai, and rewritten the sentence to “China, Vietnam, and the Philippines [95]”.

You wrote “Within the family Temoridae, there are three genera distributed in inland waters, …”. However, you mentioned later only three species, not three genera, please check. 

Reply: These three genera refer to the genus Epischura, the genus Eurytemora and the genus Heterocope, which have been found in inland waters so far.

Add the word “China” after “NE”. Please see below. “… of B. triarticuata to NE would be related …”

Reply: We have added the word “China” after “NE” (Line 351).

Add the word “China” after “NE”.

Reply: We have added the word “China” after “NE” (Line 353)

Correct the word “founded” to “found”, see below.“… three families were founded in NE China.”

Reply: We have corrected the word “founded” to “found” (Line 428).

Please correct typing errors in Ref. 15. 15. Liu, G.; Li, S. Seasonal varirations in body length and weight and ingeastion rate of Centropages tenuiremis Tompson and Scott. Acta Oceanologica Sinica 1998, 20, 104-109.

Reply: We have corrected the reference to” 15. Liu, G.; Li, S. Seasonal variations in body length and weight and ingestion rate of Centropages tenuiremis Tompson and Scott. Acta Oceanologica Sinica 1998, 20, 104-109” (Lines 488-489).

Please correct some mistakes in Ref. 88. “88. La-orsri Sanoamuang, S.W. Mongolodiaptomus mekongensis, a new species of copepod …” The correct way to write this reference is shown below: “88. Sanoamuang, L.; Watiroyram, S. Mongolodiaptomus mekongensis, a new species of copepod …”

Reply: We have corrected the reference to” 95.      Sanoamuang, L., Watiroyram, S. Mongolodiaptomus mekongensis, a new species of copepod (Copepoda, Calanoida, Diap-tomidae) from temporary waters in the floodplain of the lower Mekong River Basin. Raffles Bulletin of Zoology 2018, 66, 782-796” (Lines 660-661).

Reviewer 3 Report

Comments and Suggestions for Authors

Review for the paper "Diversity of Freshwater Calanoid Copepods (Crustacea: Copepoda: Calanoida) in Northeastern China" by Ruirui Ding, Le Liu, Shusen Shu, Yun Li, Feizhou Chen submitted to "Diversity".

General comment.

Copepods play a crucial ecological role in freshwater environments by providing a direct link between primary producers and higher trophic levels. Little research has been conducted on zooplankton in the lakes of northeastern China, and the few studies that do exist are taxonomically limited. Most of these are descriptions of copepod species or have focused on major taxa. In general, the paper is well written in terms of introduction, materials and methods, results and most of the discussion. My main concerns are mainly related to the limited interpretation of biotic-environmental interactions.

Major points.

1. Discussion. The authors must discuss the distribution patterns of copepod abundance and biomass. Comparisons of these variables from other similar ecosystems would be useful to make the study more interesting to the international readership.

2. Discussion. The authors must discuss the results of section 2.3. They must explain the mechanisms underlying copepod-environment interactions in lakes.

3. The ecological significance of the results must be indicated in the discussion.

Specific remarks.

L20. What is TP? The full form of any abbreviation should be given the first time it is used.

L19-22. The sentence is awkward. Consider replacing "Canonical Correspondence Analysis (CCA) and Redundancy Analysis (RDA) revealed that TP had the most contribution to the environmental variables that impacted the distribution of calanoid copepods, including both fresh and saline-alkaline lakes" with "Canonical Correspondence Analysis (CCA) and Redundancy Analysis (RDA) revealed that TP was the most important environmental variable that impacted the distribution of calanoid copepods from fresh and saline-alkaline lakes".

Materials and Methods. All statistical procedures must be described in the paper. When presenting the main results, the authors report the results of comparisons using ANOVA. However, the test procedures are not included in the MS. Please include the relevant description in the methods.

Figure 1. Please increase the font size in the map.

L84. Consider replacing "2.2 Sampling collection, preservation, and identification" with "2.2 Sampling, preservation, and identification".

L96-97. Consider replacing "were identified by Shen and Song [7], Sheveleva et al., 2021 [34]" with "were identified with relevant guides [7, 34]".

Section 2.3. It is unclear which biological input variables were used in RDA and CCA (abundance or biomass). Clarify.

Figure 2. Lake groups (types 1-5) need to be indicated in the dendrogram.

Fig. 3. Please increase the font size in the plot.

L163. What about differences between lake types based on zooplankton abundance?

L224. Consider replacing "right Antennule" with "right antennule".

Section 3.3. and Figure 6. The authors noted in the abstract that TP was the most contributing variable in the copepod pattern. However, other variables had a significant effect on certain copepods. These need to be mentioned in the abstract.

Section 3.3. and Figure 6. Explain why only selected copepod taxa were analyzed in RDA.

Reference list. Authors must carefully check the Latin names of species and genera. In some cases, these are in regular font instead of italics. L508. In some cases, family names are italicized instead of being in normal font (e.g., L514). In some references the authors use 'calanoida' instead of 'Calanoida'.

Reference 71. Correct the author's name. Larysa is the name, not the surname. Must be Samchyshyna L.

Comments on the Quality of English Language

Minor revision

Author Response

Dear editors and reviewer,

We feel great thanks for your professional review work on our article. As you are concerned, there are several problems that need to be addressed. According to your nice suggestions, we have made extensive corrections to our previous draft. For your suggestions, we have highlighted the corrections in the manuscript in blue. The detailed corrections are listed below.

  1. Discussion. The authors must discuss the distribution patterns of copepod abundance and biomass. Comparisons of these variables from other similar ecosystems would be useful to make the study more interesting to the international readership.

Reply: We have added the comparisons of variables from other similar ecosystems (eutrophic lakes in Southwestern China and the Yangtze River Basin region) (Lines 272-274).

  1. Discussion. The authors must discuss the results of section 2.3. They must explain the mechanisms underlying copepod-environment interactions in lakes.

Reply: We have added discussion about copepod-environment interactions (Lines 394-418).

  1. The ecological significance of the results must be indicated in the discussion.

Reply: We have added discussion about the ecological significance of the results (Lines 369-371).

Specific remarks.

L20. What is TP? The full form of any abbreviation should be given the first time it is used.

Reply: We have added the full form of TP (Line 21).

L19-22. The sentence is awkward. Consider replacing "Canonical Correspondence Analysis (CCA) and Redundancy Analysis (RDA) revealed that TP had the most contribution to the environmental variables that impacted the distribution of calanoid copepods, including both fresh and saline-alkaline lakes" with "Canonical Correspondence Analysis (CCA) and Redundancy Analysis (RDA) revealed that TP was the most important environmental variable that impacted the distribution of calanoid copepods from fresh and saline-alkaline lakes".

Reply: After combining the suggestions of the reviewers, we have rewritten the sentence to “Canonical Correspondence Analysis (CCA) and Redundancy Analysis (RDA) revealed that calanoid copepods were significantly correlated with total phosphorus (TP), total nitrogen, conductivity, nitrate nitrogen, altitude and solved organic carbon. TP was the most important environmental variable that impacted the distribution of calanoid copepods, including both fresh and saline-alkaline lakes” (Lines 19-23).

Materials and Methods. All statistical procedures must be described in the paper. When presenting the main results, the authors report the results of comparisons using ANOVA. However, the test procedures are not included in the MS. Please include the relevant description in the methods.

Reply: We have added relevant description for ANOVA (Lines 113-114).

Figure 1. Please increase the font size in the map.

Reply: We have increased the font size in Figure 1.

L84. Consider replacing "2.2 Sampling collection, preservation, and identification" with "2.2 Sampling, preservation, and identification".

Reply: We have corrected “2.2 Sampling collection, preservation, and identification” to “2.2 Sampling, preservation, and identification” (Line 86).

L96-97. Consider replacing "were identified by Shen and Song [7], Sheveleva et al., 2021 [34]" with "were identified with relevant guides [7, 34]".

Reply: After combining the suggestions of the reviewers, we have rewritten the sentence to “In the laboratory, calanoid copepods were identified using the identification keys of Shen and Song [7] and Sheveleva et al., 2021 [34]” (Lines 98-99).

Section 2.3. It is unclear which biological input variables were used in RDA and CCA (abundance or biomass). Clarify.

Reply: We used calanoids biomass for biological variables in RDA and CCA. We have added this information in Line 117.

Figure 2. Lake groups (types 1-5) need to be indicated in the dendrogram.

Reply: We have added Lake groups (types 1-5) in the dendrogram.

Fig. 3. Please increase the font size in the plot.

Reply: We have increased the font size in the plot in Figure 3.

L163. What about differences between lake types based on zooplankton abundance?

Reply: There was no significant difference between lake types based on zooplankton abundance. We have added this information in Lines 174-175.

L224. Consider replacing "right Antennule" with "right antennule".

Reply: We have corrected “right Antennule” to “right antennule” (Lines 235-236).

Section 3.3. and Figure 6. The authors noted in the abstract that TP was the most contributing variable in the copepod pattern. However, other variables had a significant effect on certain copepods. These need to be mentioned in the abstract.

Reply: We have added other variables in Lines 21-22 in the abstract.

Section 3.3. and Figure 6. Explain why only selected copepod taxa were analyzed in RDA.

Reply: RDA and CCA were selected for type 1 and type 2 lakes, and copepod taxa selected for analysis were those found in these lakes.

Reference list. Authors must carefully check the Latin names of species and genera. In some cases, these are in regular font instead of italics. L508. In some cases, family names are italicized instead of being in normal font (e.g., L514). In some references the authors use 'calanoida' instead of 'Calanoida'.

Reply: We have reconfirmed the Latin names and forms of species and genera and corrected them (Lines 559, 625, 653, and 667).

Reference 71. Correct the author's name. Larysa is the name, not the surname. Must be Samchyshyna L.

Reply: We have corrected the reference to “Larysa, S.” to “Samchyshyna, L. Copepoda calanoida of the Shatski Lakes (Ukraine). Vestnik Zoologii 2001, 35, 47-51” (Line 620).

Round 2

Reviewer 3 Report

Comments and Suggestions for Authors

No further comments.